# Inhibition of TOR in *Chlamydomonas reinhardtii* Leads to Rapid Cysteine Oxidation Reflecting Sustained Physiological Changes

**DOI:** 10.3390/cells8101171

**Published:** 2019-09-28

**Authors:** Megan M. Ford, Amanda L. Smythers, Evan W. McConnell, Sarah C. Lowery, Derrick R. J. Kolling, Leslie M. Hicks

**Affiliations:** 1Department of Chemistry, The University of North Carolina at Chapel Hill, Chapel Hill, NC 27599, USA; megan.ford@unc.edu (M.M.F.); evanmcc@live.unc.edu (E.W.M.); slowery@unc.edu (S.C.L.); 2Department of Chemistry, Marshall University, Huntington, WV 25755, USA; asmyther@email.unc.edu (A.L.S.); kolling@marshall.edu (D.R.J.K.)

**Keywords:** redox signaling, target of rapamycin, photosynthesis, proteomics, *Chlamydomonas reinhardtii*

## Abstract

The target of rapamycin (TOR) kinase is a master metabolic regulator with roles in nutritional sensing, protein translation, and autophagy. In *Chlamydomonas reinhardtii*, a unicellular green alga, TOR has been linked to the regulation of increased triacylglycerol (TAG) accumulation, suggesting that TOR or a downstream target(s) is responsible for the elusive “lipid switch” in control of increasing TAG accumulation under nutrient limitation. However, while TOR has been well characterized in mammalian systems, it is still poorly understood in photosynthetic systems, and little work has been done to show the role of oxidative signaling in TOR regulation. In this study, the TOR inhibitor AZD8055 was used to relate reversible thiol oxidation to the physiological changes seen under TOR inhibition, including increased TAG content. Using oxidized cysteine resin-assisted capture enrichment coupled with label-free quantitative proteomics, 401 proteins were determined to have significant changes in oxidation following TOR inhibition. These oxidative changes mirrored characterized physiological modifications, supporting the role of reversible thiol oxidation in TOR regulation of TAG production, protein translation, carbohydrate catabolism, and photosynthesis through the use of reversible thiol oxidation. The delineation of redox-controlled proteins under TOR inhibition provides a framework for further characterization of the TOR pathway in photosynthetic eukaryotes.

## 1. Introduction

Target of rapamycin (TOR) is a conserved Ser/Thr kinase and master regulator of cellular growth and homeostasis in eukaryotes, with significant control over nutrient-responsive pathways including macromolecular anabolism and catabolism as well as vacuole formation and autophagy [1,2,3,4,5,6,7]. In yeast, mammals, and other complex eukaryotes, TOR has been identified in two distinct complexes, TORC1 and TORC2 [2,8,9,10,11], but in photosynthetic organisms only the components of TORC1 have been identified [2,4]. While the role of TOR in mammalian species and yeast has been studied extensively [2,12,13], its role in photosynthetic organisms is less established [4,5].

Specific chemical inhibitors of TOR have been used to delineate some of the targets of the TOR pathway in mammalian cells [14], yeast [15], and photosynthetic organisms [16,17,18] through proteomics and transcriptomic analyses as well as physiological characterization. Directly inhibiting TOR results in similar phenotypic changes observed under nitrogen deprivation [19,20], including an increase in triacylglycerol (TAG) content and a decrease in protein synthesis [21]. This suggests that the eukaryotic “lipid switch”—the protein(s) responsible for upregulating lipid formation under nutrient limitation—is regulated by the TOR pathway [19,20,21,22,23]. TAG induction is a response to autophagy [24,25], a process highly regulated by TOR in algae. TOR inhibition of the model algal species *Chlamydomonas reinhardtii* (*C. reinhardtii*) induces vacuolization, cell bleaching, and production of autophagy-specific cell markers [1,26]. Additionally, TOR is involved in the regulation of protein synthesis, with the small molecule inhibitor rapamycin decreasing protein synthesis through the phosphorylation of BiP, an endoplasmic reticulum chaperone and member of the HSP70 superfamily involved in post-translational protein folding [27]. Furthermore, TOR has a significant role in the overall nutrient metabolism of the cell, including the aforementioned TAG synthesis pathways as well as the tricarboxylic acid cycle, which is downregulated under TOR inhibition and decreases carbohydrate catabolism [18,28].

In addition to phosphorylation, protein oxidation can play an important role in the regulation of stress response, as studied previously using exogenous H_2_O_2_ in algae [29]. However, little work has been done to look at the role of oxidative signaling in response to the TOR pathway, either through regulation of the TOR pathway via oxidation or the TOR pathway controlling oxidative stress response in the cell. Oxidative stress can cause increased protein oxidation on the reactive thiol groups of cysteines (Cys) in the form of disulfide bonds, S-glutathionylation, S-nitrosylation, and S-sulfenylation, all of which can leverage a reversible regulatory mechanism. TOR itself has been shown in mammalian cells to be regulated by the reversible formation of disulfide bonds by thioredoxin-1 [30]. In photosynthetic organisms, reversible oxidation sites have been previously identified on enzymes involved in signaling, stress response, transcription and translational control, and metabolism [22,31,32,33,34,35]; all of these are pathways also known to be regulated by TOR, indicating that TOR is likely utilizing reversible oxidative signaling. While many targets of the TOR pathway have known sites of reversible oxidation, the extent to which reversible oxidation is implemented in TOR signaling is unknown, as are the resulting physiological responses.

Herein, quantitative proteomics of TOR inhibition in *C. reinhardtii* via enrichment of reversibly oxidized Cys (Figure 1) reveals significant increases in reversible oxidation throughout the proteome over the first hour of inhibition, including sites related to lipid synthesis, carbohydrate synthesis and catabolism, and photosynthesis. Physiological changes measured up to 48 h, including increases in TAG and carbohydrate content, correlate with oxidative changes, delineating the impact of TOR inhibition on changing cell metabolism. Additionally, inhibition of photosynthesis in response to TOR inhibition was characterized for the first time. This analysis demonstrates the overlap of physiological control and signaling regulation of the TOR pathway through reversible oxidation of thiols.

## 2. Materials and Methods

### 2.1. Strain, Culture Growth and Treatment Conditions

Wild-type *C. reinhardtii* strain CC-2895 6145c mt- and Hutner’s trace elements were purchased from the Chlamydomonas Resource Center (St. Paul, MN, USA) and batch cultures were maintained photoheterotrophically on Tris-acetate-phosphate (TAP) agar plates. For physiological experiments, *C. reinhardtii* was inoculated into 25 mL of TAP medium using a 250 μL inoculum in a 50 mL Erlenmeyer flask top capped with aluminum foil. For proteomic experiments, cells were inoculated into 250 mL of TAP liquid medium [36] in 500 mL sterile Erlenmeyer flasks. Cultures were grown photoheterotrophically in quadruplicate, using a 2.5 mL inoculum from a mid-exponential-phase (OD_750 nm_ 0.4–0.5) culture and grown under constant white-light conditions of 30 µmol photons · m^−2^ · s^−1^ at 25 °C and at an orbital rotational speed of 120 rpm on a VWR International model 1000 standard orbital shaker (Radnor, PA, USA).

AZD8055 (MedChem Express; Monmouth Junction, NJ, USA) dissolved in dimethyl sulfoxide (DMSO; Fisher Scientific, Waltham, MA, USA) was added when cells reached an OD_750 nm_ of 0.4 ± 0.1 to the saturating concentration of 700 nM [16]. Control cultures were given an equal volume of DMSO without AZD8055. For physiological measurements, the cells were harvested immediately after dosing and then every 12 h through 48 h of treatment (Appendix A). For proteomics experiments, the cells were harvested immediately after dosing as well as 15 min, 30 min, and 1 h post-dosing (Appendix A). Cells were harvested by centrifuging for 2 min at 3220× *g* and discarding the supernatant. Cell pellets were flash-frozen using liquid nitrogen and stored at −80 °C until use.

### 2.2. Spectroscopic Cell Density and Cell Diameter

Spectroscopic cell density (turbidity) was measured using a Shimadzu UV-1800 spectrophotometer (Shimadzu Corp., Kyoto, Japan) at 750 nm as previously described [37,38]. Cell diameter was determined using a micrometer slide on a VistaVision light microscope (VWR International, at 1000× magnification. FIJI software was used for image analysis [39].

### 2.3. Pigment Extraction

Pigments were extracted as previously described and measured from 470 to 700 nm [37]. Chlorophyll *a* content (Chl *a*) was calculated using the following equation [40]: [Chl *a*] = (12.47 × Abs_665.1_) − (3.62 × Abs_649.1_).

### 2.4. Cell Dry Weight Measurement

Dry mass was measured as previously reported [38]. Briefly, 1 mL of cells were pelleted and rinsed with H_2_O and filtered onto pre-weighed 1 μm, 25 mm GF/B Whatman glass microfiber filters (Whatman International Ltd., Maidstone, UK) using a Büchner funnel. Filters and cells were dried in an incubator at 75 °C for 24 h before being weighed on a Secura 125-1S analytical balance (Sartorius, Göttingen, Germany).

### 2.5. Lipid Analysis

Lipid extractions were performed as previously described using a modified methyl *tert*-butyl ether (MTBE) extraction [41]. A 10 mL sample was pelleted and the supernatant discarded. Cell pellets were lysed with 1 mL of methanol (Fisher Scientific) and incubated in a 9 mL tube with 4 mL of MTBE (Fisher Scientific) for 1 h before adding 1 mL H_2_O and incubating for another 15 min. Suspensions were centrifuged for 15 min at 10,000× *g* and the organic layer was removed by a Pasteur pipette into a pre-weighed 4 cm tube and dried under vacuum. The extraction was completed twice to ensure near-complete recovery of lipid mass. Tubes were weighed on a 125-1S Secura analytical balance. Neutral lipids were measured using Nile Red (Sigma-Aldrich, St. Louis, MO, USA) fluorescent staining [42]. Cells were incubated in the dark for 10 min following a 1:1 dilution in 2 μg · mL^−1^ Nile Red in DMSO. Fluorescence was measured using a SpectraMax M2 (Molecular Devices, LLC, San Jose, CA, USA) with a nine-point well scan and an excitation wavelength of 530 nm and emission wavelength of 580 nm. 

### 2.6. Biochemical Composition

Terminal carbohydrates were assayed as previously described using the acid-phenol assay [38,43]. Briefly, 100 µL of sample was collected in triplicate from each culture and pelleted, discarding the supernatant. The pellet was then resuspended with 100 µL H_2_O before adding 500 µL concentrated H_2_SO_4_ (Fisher Scientific) and vortexing. After a 15 min incubation at room temperature (RT), 100 µL of 5% (*w*/*v*) phenol (Fisher Scientific) in H_2_O was added and vortexed. After 15 min, the absorbance of each sample was measured at 490 nm using a Shimadzu UV-1800 spectrophotometer. Calibration curves were prepared daily using a freshly prepared 0.05 mg/mL D-glucose (Sigma-Aldrich) stock solution.

Terminal proteins were extracted following a previously described method [44] and were assayed using a modified Lowry assay [45,46]. A stock of Lowry Reagent D was prepared daily in a 48:1:1 ratio of Lowry Reagents A (2% *w*/*v* Na_2_CO_3_ in 0.1N NaOH; Fisher Scientific), B (1% *w*/*v* NaK tartrate; Fisher Scientific), and C (0.5% *w*/*v* CuSO_4_·5H_2_O; Fisher Scientific) and the Folin-Ciocalteu reagent (Sigma-Aldrich) was prepared daily with a 1:1 ratio of H_2_O. All biological replicates were measured in triplicate by adding 50 µL of protein extract to 950 µL of Lowry Reagent D before vortexing and incubating at RT for 10 min. Following incubation, 100 µL of diluted Folin-Ciocalteu reagent was added before thoroughly vortexing and incubating at RT for 30 min. The absorbance of each sample was measured at 600 nm using a Shimadzu UV-1800 spectrophotometer and quantified daily using a five-point calibration curve prepared from a 2 mg · mL^−1^ bovine serum albumin stock solution (Fisher Scientific).

### 2.7. Chlorophyll Fluorescence Induction in Vivo

The Chl *a* OJIP transient is a highly sensitive measurement of photosynthesis that is used to infer information about the efficiency of electron transport through photosystem II (PSII) [47]. When a dark-adapted phototrophic sample is exposed to actinic light, the Chl *a* fluorescence emits in a polyphasic rise with four characteristic ‘steps,’ O, J, I, and P. The O step corresponds with the origin, or minimal fluorescence, the J and I are for the inflections at 2 and 30 ms, respectively, and the P is the maximum fluorescence output. Using the O and P steps, it is possible to calculate the *F_V_/F_M_*, in which *F_V_* denotes the variable fluorescence calculated by taking the difference between *F_M_* and *F_O_* and *F_M_* is the fluorescence at the P step (Appendix A). *F_V_*/*F_M_* is a measure of the maximum quantum yield of primary photochemistry in a dark-adapted state and is frequently used to express overall photosynthetic efficiency. The steps of the OJIP transient have also been shown to correspond to the oxidation state of the plastoquinone pool; as the steps increase in intensity, the overall oxidation state of the pool also increases [48]. 

Photosynthetic electron transfer fluxes were inferred from Chl *a* fluorescence using a Photon Systems Instruments FL 3500 fluorometer (Drasov, Czech Republic) as previously described [38]. The OJIP protocol included a 1s actinic illumination using a 630 nm light at an intensity of 2400 µmole photons · m^−2^ · s^−1^. Fluorometry OJIP parameters (JIP test) were calculated as outlined by Stirbet [47].

Additionally, Chl *a* fluorescence was used to determine the proportion of active reducing centers, as using two subsequent actinic pulses separated by 1 s of darkness allows further information regarding the redox state of the Q_B_ reducing centers of PSII. While the first pulse was conducted following dark adaptation, meaning that all the reaction centers were open, the second pulse only excited so-called ‘fast-opening’ reaction centers, allowing for the calculation of non-reducing centers (centers which are unable to open in time for the second pulse) through the equation:$$B_{0}=\left(F_{V}/F_{M}-F_{V}{}^{*}/F_{M}{}^{*}\right)/F_{V}/F_{M},$$
where *F_V_/F_M_* is derived from the first pulse and *F_V_*/F_M_** is derived from the second pulse. 

### 2.8. Pulse Amplitude Modulated (PAM) Fluorescence

When photons (or excitons) reach PSII reaction centers, they have one of three fates: they may be used for photochemistry, emitted as fluorescence, or dissipated as heat via an internal conversion phenomenon called non-photochemical quenching (NPQ). These three fates combine to be unity, meaning that a change in the abundance of one will result in proportional changes in the others. Thus by determining the amount of NPQ and photochemistry through fluorescence techniques, a total picture of photon fate can be generated [49]. 

To measure NPQ, a quenching analysis of PAM fluorescence was used on dark-adapted cells with an actinic intensity of 300 µmol photons · m^−2^ · s^−1^, a saturating pulse intensity of 64,000 µmol photons · m^−2^ · s^−1^, and a measuring flash voltage of 80%. There was a dark relaxation duration of 20 s between pulses [50]. Photochemical coefficients were calculated as previously reported [48]. 

### 2.9. Protein Extraction for Proteomics Analyses

Frozen cell pellets (0.3 g FW from 50 mL of culture) were lysed in 10 mL of phosphate-buffered saline (PBS) with 0.5% SDS (Sigma-Aldrich), 0.1% Triton X-100 (Sigma-Aldrich), and ¼ tablet of cOmplete, EDTA-free protease inhibitor cocktail (Roche, Basel, Switzerland). Reduced Cys were blocked using 100 mM *N*-ethylmaleimide (NEM; Sigma-Aldrich) by adding 1 mL of 1 M NEM dissolved in 50% ethanol (Fisher Scientific). The reaction was incubated for 2 h at RT protected from light before centrifuging for 5 min at 3220× *g* and 4 °C to form a white pellet of cell debris. The supernatant was added to 10 mL of cold acetone (Fisher Scientific) and incubated for 30 min at −20 °C before centrifuging to pellet proteins. Samples were resuspended in 10 mL of PBS with 0.25% SDS and 4 M urea (Sigma-Aldrich) by aspirating back and forth with a 1 mL pipette tip. Protein concentration was estimated using the CB-X Protein Assay (G-Biosciences, St. Louis, MO, USA) and normalized to 1 mg/mL with resuspension buffer. Aliquots were taken for global proteomic (100 µg) and oxidized Cys enrichment analysis (1 mg).

### 2.10. Global Proteomics

Sample lysates (100 µg) were incubated on a covered ThermoMixer (Eppendorf, Hamburg, Germany) set to 25 °C and 1000 rpm. Disulfide bonds were reduced with 10 mM dithiothreitol (DTT; Sigma-Aldrich) for 30 min before directly adding 30 mM NEM for 30 min to alkylate Cys residues. Samples were mixed with 1 mL of cold acetone to precipitate proteins and centrifuged for 5 min at 10,000× *g* and 4 °C. Pellets were resuspended (500 µL) in 50 mM Tris, pH 8 with 2 M urea and digested with 2.5 µg of Trypsin Gold (Promega, Madison, WI, USA) overnight (> 16 h). The digestion was quenched (20 µL) with 5% trifluoroacetic acid (TFA; Fisher Scientific) and desalted with solid-phase extraction (SPE).

### 2.11. Oxidized Cys Enrichment

Reversible oxidation changes were measured using an oxidized Cys resin-assisted capture enrichment strategy, abbreviated as OxRAC, that has been described previously (Figure 1) [35,51]. Briefly, protein lysates (1 mg) were incubated with 10 mM DTT for 1 h at RT to reduce all reversibly oxidized Cys before precipitating proteins with 10 mL of cold acetone. Samples were incubated for 30 min at −20 °C before centrifuging for 5 min at 3220× *g* and 4 °C to collect proteins. Samples were resuspended in 1 mL of 50 mM Tris, pH 8 with 0.5% SDS and 4 M urea by aspirating back and forth with a 1 mL pipette.

Thiopropyl Sepharose 6B (TPS6B; GE Healthcare, Pittsburgh, PA, USA) resin was rehydrated in water and washed with 50 mM Tris, pH 8 before suspending to a 100 mg/mL slurry. Each sample was mixed with 50 mg of TPS6B resin (0.5 mL slurry) and incubated with end-over-end rotation for 2 h to covalently enrich proteins with reduced Cys. Samples were transferred to a MobiSpin column (Boca Scientific, Westwood, MA, USA) and nonspecifically bound proteins were removed by washing the resin (400 µL each) in 50 mM Tris, pH 8 with 0.5% SDS, 50 mM Tris, pH 8 with 2 M NaCl (Sigma-Aldrich), 80% acetonitrile (Fisher Scientific) with 0.1% TFA, and 50 mM Tris, pH 8. On-resin digestion of Cys-bound proteins was performed in 250 µL of 50 mM Tris, pH 8 with 2.5 µg of Trypsin Gold (Promega) and incubated overnight (> 16 h) with agitation at RT. The unbound peptide flow-through was separated from the Cys-bound peptides by briefly centrifuging the spin columns. Samples were washed (400 µL) using 50% acetonitrile and subsequently water. Bound Cys-containing peptides were eluted from the resin using 50 mM DTT (250 µL) for 15 min with agitation at RT and centrifuged to collect. The resin was washed twice with 50% acetonitrile (200 µL) and collected with the eluate. Samples were dried by vacuum centrifugation and SPE desalted as described below.

### 2.12. Solid-Phase Extraction

Desalting of samples was performed using 50 mg/1.0 mL Sep-Pak C18 cartridges (Waters, Milford, MA, USA) held in a SPE 24-position vacuum manifold (Phenomenex, Torrance, CA, USA) at a maximum flow rate of 1 drop/s. Resin was first pre-eluted using 1 mL of 80% acetonitrile with 0.1% TFA before equilibration with 1 mL of water with 0.1% TFA. Samples were acidified to pH 3 using 5% TFA and loaded onto the cartridges in two passes before washing with 1 mL of water with 0.1% TFA. Peptides were eluted using 1 mL of 80% acetonitrile with 0.1% TFA and dried by vacuum centrifugation.

### 2.13. LC-MS/MS Analysis

Samples were analyzed using a nanoAcquity UPLC (Waters) coupled to a TripleTOF 5600 mass spectrometer (AB Sciex, Framingham, MA, USA). Mobile phase A consisted of water with 0.1% formic acid (Fisher Scientific) and mobile phase B was acetonitrile with 0.1% formic acid. Injections (5 µL) were made to a Symmetry C_18_ trap column (100 Å, 5 µm, 180 µm x 20 mm; Waters) with a flow rate of 5 µL/min for 3 min using 99% A and 1% B. Peptides were then separated on a HSS T3 C_18_ column (100 Å, 1.8 µm, 75 µm x 250 mm; Waters) using a linear gradient of increasing mobile phase B at a flow rate of 300 nL/min. Mobile phase B increased from 5% to 35% in 90 min before ramping to 85% in 5 min, where it was held for 5 min before returning to 5% in 2 min and re-equilibrating for 13 min. The mass spectrometer was operated in positive polarity and the NanoSpray III source had ion source gas 1 set to 15, curtain gas at 25, IonSpray voltage floating at 2400, and interface heater temperature at 150. MS survey scans were accumulated across *m*/*z* range of 350–1600 for 250 ms. For data-dependent acquisition, the mass spectrometer was set to automatically switch between MS and MS/MS experiments for the first 20 features above 150 counts having +2 to +5 charge state. Precursor ions were fragmented using rolling collision energy and accumulated in high sensitivity mode across *m*/*z* range 100–1800 for 85 ms. Dynamic exclusion for precursor *m*/*z* was set to 8 s. Automatic calibration was performed every 8 h using a tryptic digest of BSA protein standard (Thermo Scientific) to maintain high mass accuracy in both MS and MS/MS acquisition.

### 2.14. Database Searching and Label-Free Quantification

Acquired spectral files (*.wiff) were imported into Progenesis QI for proteomics (Nonlinear Dynamics, version 2.0; Northumberland, UK). Peak picking sensitivity was set to maximum of five and a reference spectrum was automatically assigned. Total ion chromatograms (TICs) were then aligned to minimize run-to-run differences in peak retention time. Each sample received a unique factor to normalize all peak abundance values resulting from systematic experimental variation. Alignment was validated (> 80% score) and a combined peak list (*.mgf) was exported for peptide sequence determination and protein inference by Mascot (Matrix Science, version 2.5.1; Boston, MA, USA). Database searching was performed against the *Chlamydomonas reinhardtii* UniProt database (https://www.uniprot.org/proteomes/UP000006906, 18,828 canonical entries) with sequences for common laboratory contaminants (https://www.thegpm.org/cRAP/, 116 entries) appended. Searches of MS/MS data used a trypsin protease specificity with the possibility of two missed cleavages, peptide/fragment mass tolerances of 15 ppm/0.1 Da, and variable modifications of protein *N*-terminus acetylation, and methionine oxidation. Alkylation of Cys with NEM (+125.0477 Da, C_6_H_7_NO_2_) was set as a fixed modification for global proteomic samples and variable for oxidized Cys enrichments. Significant peptide identifications above the identity or homology threshold were adjusted to less than 1% peptide FDR using the embedded Percolator algorithm [52] and imported to Progenesis for peak matching. Identifications with a Mascot score less than 13 were removed from consideration in Progenesis before exporting both “Peptide Measurements” and “Protein Measurements” from the “Review Proteins” stage.

### 2.15. Data Analysis and Statistics

For physiological measurements, the data were analyzed through multiple comparisons of means conducted using Welch’s t-tests. The family-wise error rate for each figure was maintained at 0.05 through the use of the Holm-Bonferroni method, unless stated otherwise. To determine statistical significance of changes over time, data was analyzed through one-way repeated measures analysis of variance (ANOVA) conducted with Graphpad Prism (Graphpad Software, v7.01; San Diego, CA, USA). Statistical significance is indicated numerically through increasing asterisks, where * indicates *p* ≤ 0.05, ** indicates *p* ≤ 0.01, *** indicates *p* ≤ 0.005, and **** indicates *p* ≤ 0.001. Figures show the means of quadruplicate data and the error bars denote the standard error of the measurement.

For LC-MS/MS-based proteomics, data were parsed using custom scripts written in R for pre-processing and statistical analysis (https://github.com/hickslab/QuantifyR).

For global proteomic analysis, leading protein accessions were considered from the “Protein Measurements” data and kept if there were ≥ 2 shared peptides and ≥ 1 unique peptide assigned. Proteins were removed if there was not at least one condition with 3/4 nonzero values across the Progenesis-normalized abundance columns. Values were log_2_-transformed and we applied a conditional imputation strategy using the imp4p package [53], where conditions with at least one nonzero value had missing values imputed using the *impute.rand* function with default parameters. For cases where a condition had only missing values, the *impute.pa* function was used to impute small numbers centered on the lower 2.5% of values in each replicate. Statistical significance was determined using a two-tailed, equal variance *t*-test and the method of Benjamini and Hochberg (BH) was used to correct *p*-values for multiple comparisons [54]. Fold change was calculated by the difference of the mean abundance values between conditions being compared. Only observations with FDR-adjusted *p* < 0.05 and log_2_-transformed fold change +/− 1.5 were considered significantly different.

For the OxRAC experiment, we summarized the “Peptide Measurements” data, which contains peak features with distinct precursor mass and retention time coordinates matched with a peptide sequence identification from the database search results.

Some features were duplicated and matched with peptides having identical sequence, modifications, and score, but alternate protein accessions. These groups were reduced to satisfy the principle of parsimony and represented by the protein accession with the highest number of unique peptides found in the “Protein Measurements” data for this experiment, else the protein with the largest confidence score assigned by Progenesis. Some features were also duplicated with differing peptide identifications and were reduced to just the peptide with the highest Mascot ion score.

Results were then filtered for reversibly oxidized Cys-peptides only, defined here by the absence of NEM modification on at least one Cys residue in the peptide sequence. An identifier was created by joining the protein accession of each peptide to the particular site(s) of modification in the protein sequence. Each dataset was reduced to unique identifiers by summing the abundance of all contributing peak features (i.e., different peptide charge states, missed cleavages, and combinations of additional variable modifications). Identifiers were represented by the peptide with the highest Mascot score in each group.

Identifiers were removed if there was not at least one condition with 3/4 nonzero values across the Progenesis-normalized abundance columns. Values were log_2_-transformed and we applied the same conditional imputation strategy as used for the global proteomic analysis. 

Statistical significance was determined using one-way analysis of variance (ANOVA) and *p*-values were BH-corrected. Only observations with FDR-adjusted *p* < 0.05 and log_2_-transformed fold change +/− 2 in the 60 min condition relative to the 0 min control were considered significantly different. Unsupervised hierarchical clustering was performed on significantly different identifiers to group together similarly changing abundance trends across conditions (i.e., with time). Gene ontology (GO) annotations were pulled from UniProt and summarized/visualized for each cluster.

### 2.16. Data Availability

The mass spectrometry proteomics data have been deposited to the ProteomeXchange Consortium via the PRIDE partner repository [55] and can be accessed with the identifier PXD014819.

## 3. Results

### 3.1. Cell Growth

Cell growth of both control and TOR inhibitor-treated cultures increased by 142 and 68%, respectively, from the point of dosage until 48 h (Figure 2a). However, treatment with AZD8055 inhibited the amount of overall growth when compared to the control, with a final turbidity 32% less than the control cultures, showing that while AZD8055 treatment did not completely result in the stagnation of cell density, it did lead to a severe decrease in rate of cell growth. Dry mass measurements were congruent with the OD_750 nm_ measurements (Figure 2b). While the optical density of AZD8055-exposed cells was increasing, the cell size also significantly increased over time, whereas the cell size of control cultures did not change significantly (Figure 2c). This suggests that the increasing optical density of AZD8055-treated cultures is due to an increase in cell size rather than cell number, which is further supported by cell counting (Appendix A). Additionally, while Chl *a* in AZD8055-treated cells was significantly less than in non-inhibited cells, this accumulation did not change significantly in treated cells over time, indicating that the cells are not chlorotic - the penultimate step in photoautotrophic autophagy (Figure 2d) [56]. 

Rather, it appears as though Chl *a* synthesis is inhibited without a subsequent increase in chlorophyll degradation. Thus, cell division of AZD8055-dosed cells was significantly inhibited in comparison with the control cultures, but cell death does not appear to be initiated as a result of TOR inhibition.

### 3.2. Bulk Cell Composition 

Cells were assayed for protein, lipid, and carbohydrate content every 12 h (Figure 3a–c). No differences between control and AZD8055-treatment were observed in total protein or total lipids, with the overall compositional percent staying within error throughout the time points measured (Figure 3a,b). However, neutral lipids had significant increases in AZD8055-exposed cells, with 2.6x the control at 48 h (Figure 3d). 

Carbohydrates also significantly increased by 36 h post-dosage, with AZD8055-treated cultures accumulating 2.4x the carbohydrates vs. control cultures when normalized to dry mass (Figure 3c).

This is likely due to an increase in starch accumulation, as previous work has shown TOR-inhibition to favor lipid cycling to storage macromolecules [57]. This also could explain the increase in AZD8055-treated cell size, as cells with higher accumulations of starch are likely to have higher water accumulation that could increase the overall cell diameter [58,59].

### 3.3. Photosynthetic Output 

In order to determine if TOR inhibition affects photosynthetic productivity, Chl *a* (OJIP) fluorescence and PAM fluorescence were used in vivo following AZD8055 treatment (Figure 4). Analysis of the OJIP transient following TOR inhibition revealed a decrease in PI_abs_, the performance index on a per absorption basis, to 47% of its pre-AZD8055 productivity after just 1 h of exposure, showing that TOR has a significant and quick effect on photosynthetic electron transport (Figure 4a). Over time, this decrease in photosynthetic activity becomes more pronounced. The decrease in F_V_/F_M_ (which shows that PSII is affected by AZD8055 even in a dark-adapted state) when compared to the control paired with the increase in the overall amplitude of the OJ phase of the transient indicates that the decline in photosynthetic efficiency is the result of a decrease in the overall reduction of Q_A_ centers in PSII [60] (Figure 4b). Furthermore, the averaged trapped energy flux per PSII reaction center (TR_0_/RC), the relative number of photons absorbed through the antenna that are trapped by PSII reaction centers, decreases to 60% of the control by 36 h post-dosage (Appendix A). 

Additionally, the two-pulse method of collecting OJIP traces indicated that the AZD8055-treated cultures decreased in reduction capacity (when compared to the control) only at 12 and 24 h after treatment, after which it was not statistically different from the control (Appendix A). This shows that the lack of electron flux was not a result of damage to the acceptor side of PSII, as the increase in B_0_ does not explain the continuing decrease in photoactivity following the 24 h measurement [61]. However, while PAM fluorescence indicated that the electron flow through PSII (YII) was significantly diminished with a decrease of 1.8x after 1 h of AZD8055 treatment, the NPQ parameter was not increased (Figure 4c,d). As all absorbed photons must be accounted for through photochemistry, NPQ, or fluorescence, the decrease in photochemical flow in this case must be reallocated to fluorescence, showing a decrease in available PSII to further reduce the plastoquinone pool.

However, the M_0_ parameter of the OJIP analysis, representative of the relative rate of primary Q_A_ reduction, increases over the first hour of AZD8055 inhibition, suggesting that the plastoquinone pool is being reduced more rapidly after TOR inhibition (Figure 4a). An increase in fluorescence paired with the increase in M_0_ suggests that the plastoquinone pool is either not being sufficiently oxidized downstream or being re-reduced via cyclic electron transport, decreasing the availability of plastoquinone for the Q_A_ site of PSII and increasing the likelihood of charge recombination and subsequent fluorescence.

### 3.4. Coverage and Differential Analysis of the Reversibly Oxidized Thiol Proteome Upon TOR Inhibition

A total of 16 *C. reinhardtii* cultures, four biological replicates for each time point, were grown to mid-exponential phase before treatment. Previous TOR inhibition studies have shown that changes in signaling occur rapidly after treatment, in as little as two min [62]. Previous work in both *C. reinhardtii* and *Arabidopsis thaliana* has shown that after treatment with H_2_O_2_, changes in reversible oxidation are seen in as little as ten min [34,63]. Given these documented rapid changes and to minimize the impact of protein turnover in the experiment, time points of 0, 15, 30, and 60 min were selected to assess reversible oxidation upon TOR inhibition. Global differential proteomic analysis was performed between the 0- and 60-min time points to confirm the absence of protein turnover. Of the 1346 proteins identified across these samples (Appendix A), only one (A0A2K3DLA1, Ubiquinol oxidase) was shown to be significantly changing in abundance. 732 of these proteins identified in this global analysis, or 54%, were also measured in the oxidized Cys analysis. Thus, the observed changes in oxidation of the identified Cys sites in the time course can be confidently assessed and with little to no false positives resulting from protein turnover or expression changes. 

Identification and site-specific quantification of reversible thiol oxidation has been readily performed via differential alkylation-based methods utilizing thiol-disulfide exchange chromatography [64,65] that have been tailored for specific modifications including S-nitrosylation [66,67], S-glutathionylation [68,69], and S-acylation [70]. By using DTT rather than a modification-specific reductant, it is possible to map the entirety of the cellular redox proteome. Herein, we have applied this strategy for quantitative profiling of reversible Cys oxidation using OxRAC in the *C. reinhardtii* proteome following *in situ* TOR inhibition with exogenous AZD8055.

Cys reactivity was quenched during cell lysis under denaturing conditions in the presence of NEM to block reduced thiols, which has been shown to be rapid and efficient in alkylation [71,72]. All reversibly oxidized Cys-residues were later reduced by DTT and nascent thiols enriched at the protein-level using Thiopropyl Sepharose 6B (TPS6B) resin. On-resin trypsin digestion of Cys-bound proteins was performed and unbound peptides were washed away. Cys-bound peptides were eluted from the resin and analyzed by LC-MS/MS.

Overall, 5177 unique oxidized Cys sites were identified, quantified by 4755 peptides, referred to as identifiers, from 2234 proteins (Appendix A). Most of these peptides had only one modification site (85% of 4755), which was most likely due to Cys being particularly rare. Most proteins in the *C. reinhardtii* proteome have Cys residues (93%, 17571/18828), but there are relatively fewer Cys compared to other amino acids (i.e., on average 1.6% Cys per protein compared to 11% Gly and 16% Ala), making peptides with multiple Cys residues a less frequent occurrence. Nearly half the proteins identified had only one modified site (48% of 2234).

These sites were compared with literature where specific redox modifications were targeted in *C. reinhardtii* including S-glutahionylation [32], S-nitrosylation [73], and thioredoxin-dependent reduction [74]. Of the proteins identified in this dataset, 22 were shown previously to be S-glutathionylated, 100 were S-nitrosylated, and 188 were thioredoxin-dependent.

The most modified protein had 41 oxidized Cys sites quantified by 36 identifiers and was a large (870 residues), predicted protein (A8JFZ2_CHLRE) localized to the Golgi membrane (GO:0000139) with a Cys-rich repeat (PF00839) known to form intra-chain disulfide bonds [75]. While most Cys sites on this protein were unchanged during the treatment, one Cys identifier (C632–646) increased 2.3-fold by 60 min and could provide mechanistic insight to this otherwise uncharacterized protein.

To assess overall change in oxidation across all four time points, a one-way ANOVA test was performed. After FDR-correction, 510 identifiers from 401 proteins had a significant 0–60 min fold change, with 135 identifiers decreasing in oxidation (highlighted in blue in Appendix A) and 375 identifiers increasing in oxidation (highlighted in red in Appendix A) following TOR inhibition. A *p* < 0.05 after FDR correction and at least a two-fold change in oxidation was needed for an identifier to be considered significantly changing. Hierarchical clustering was performed on the identifiers that were significantly changing with AZD8055 treatment (Figure 5a). These identifiers, when sorted into two clusters, separate based on those identifiers that generally increase in oxidation (361 identifiers), and those that decrease (149 identifiers), suggesting the most pronounced change in oxidative state occurs after 60 min of treatment. However, there are identifiers within these data that, when a four-cluster analysis is used, show that some Cys sites have a maximum fold change before 60 min (Appendix A). This implies either a recovery of the oxidation state prior to treatment, or oxidation of the site to an extent that it becomes irreversibly oxidized. GO analysis revealed that proteins within cluster A from the two-cluster analysis, which are generally increasing in oxidation, include many physiologically important processes including translation, photosynthesis, and transcription (Figure 5b), as well as proteins with inorganic binding sites such as metal ion and ATP-binding enzymes. Many proteins in cluster B, which generally decrease in reversible oxidation, are involved with cell redox homeostasis. This finding suggests that these proteins may be highly reactive to oxidation under TOR inhibition, and rather than seeing a decrease in reversible oxidation through reduction, these Cys residues instead might be hyper-oxidized to an irreversible sulfinyl or sulfonyl modification.

## 4. Discussion

While previous research has shown the connection between TOR inhibition, lipid accumulation, and the similar phenotypic response seen with H_2_O_2_ treatment [29], this study indicates that large-scale reversible oxidative signaling is part of TOR pathway regulation and impacts all major aspects of metabolism. By investigating the 401 proteins with statistically significant changes in oxidation, focusing specifically on the proteins involved in lipid synthesis, protein translation, carbohydrate metabolism, the TOR pathway, and photosynthesis, it was possible to integrate observed physiological changes with the oxidized Cys-containing peptides, providing a framework for determining how TOR impacts the changing phenotype of *C. reinhardtii*. 

### 4.1. Lipid Metabolism

The physiological response following TOR inhibition shows an increase in neutral lipids while overall lipid content remains steady. These data, when paired with the increase in cell size of AZD8055-treated cultures, is indicative of carbon reallocation in which the cells are redistributing lipids from phospholipids into TAGs for long-term storage—a phenomenon that has been characterized in *C. reinhardtii* following nitrogen deprivation [20]. These results are supported by the regulation of 48 lipid-related proteins through changes in reversible oxidation upon TOR inhibition (Appendix A). Of the 106 identifiers, eight were seen to significantly increase in oxidation upon inhibition including Cys302 on glycerol-3-phosphate acyltransferase (H9CTH0, FC: 49.67), Cys35-Cys39-Cys44 on phospholipase A2 (A8I2I2, FC: 9.59), Cys387-Cys390 on chloroplast ω6 desaturase (O48663, FC: 5.82), Cys653 on phospholipase B-like (A0A2K3DXV3, FC: 4.86), and C278 on phosphoglycerate kinase (A8JC04, FC: 2.74). Both phospholipases are involved in the cleavage of fatty acids in phospholipids, hydrolyzing the major component of the cell membrane. It has been previously shown that under nitrogen deprivation there is an increase in phospholipase abundance and other components responsible for membrane remodeling, reflecting a similar response in regulation between these two stressors [20]. Phospholipases are also known to be Cys-rich, with many of these Cys residues involved in disulfide bonds [76]. While the impact of these oxidation sites is unknown, previous work has shown that oxidized Cys in mammalian cells regulate the activity of phospholipases [76]. Conversely, chloroplast ω6 desaturase is involved in lipid synthesis, introducing a double bond in the biosynthesis of 16:3 and 18:3 fatty acids, an important component of plant membranes [77]. The diversion of lipids from membrane components to TAGs suggests that this oxidation site may regulate the activity of this desaturase. Phosphoglycerate kinase (PGK) is involved in glycolysis and in carbon metabolism, catalyzing the reaction of 1,3-bisphosphoglycerate into 3-phosphoglycerate in glycolysis and the reverse reaction in the Calvin-Benson-Bassham Cycle (CBBC) [78]. This enzyme is known to be highly regulated by oxidation, with previous work in cyanobacteria showing that the conserved Cys residue found in this study plays an important role in regulating the activity of this kinase [78]. This previous work showed that oxidation of this Cys greatly diminished the activity of PGK, suggesting that the increase in oxidation seen in this study could be inhibiting the kinase, demonstrating regulation of glycolysis/CBBC by the TOR pathway via reversible oxidation [78].

### 4.2. Protein Translation

Although there was not a significant physiological change in total protein due to treatment, a number of translation-related proteins were identified as being regulated by reversible oxidation upon TOR inhibition (Appendix A). The latter better correlates to previous research studying physiological changes in *C. reinhardtii* following TOR inhibition [27]. A total of 204 identifiers were quantified on 101 unique proteins involved in translation. Of these identifiers, 17 were found to increase in oxidation, while three significantly decreased. One of these identifiers is Cys82 on elongation factor Tu (P17746, FC: 3.23), a protein that promotes binding of aminoacyl-tRNA to the ribosomal A-site during protein synthesis. This protein has been shown to be oxidized in a number of different bacteria [79,80,81], and although the exact function of this oxidation is unknown, the conservation across multiple bacterial strains as well as in *C. reinhardtii* suggests this post-translational modification may be important for regulation of its activity. 

Many ribosomal proteins were shown to contain reversible oxidation sites, including 121 identifiers on 62 proteins, nine of which significantly increased and two that significantly decreased (Appendix A). Ribosomal proteins in yeast have been shown to be regulated by reversible oxidation with many of these proteins containing a conserved CX_2_C-X_9-47_-CX_2,4_C site where oxidation occurs on one of these Cys residues [82]. These conserved motifs have been shown to be Zn^+2^ binding, stabilizing in protein folding, and their activity is redox-regulated. Two of the proteins in this study, ribosomal protein L37a (A8HY08) and ribosomal protein L36a (A8IM74), have this conserved motif. Ribosomal protein L37a demonstrated reversible oxidation on Cys39, the first Cys in the conserved motif, although it was not shown to be significantly changing. Ribosomal protein L36a had significantly changing oxidation on Cys86 (FC: 2.05), a Cys outside the conserved motif. Ribosomal protein L10 (A8IZK3) also has an identifier, Cys140, that significantly increases 2.07-fold in oxidation upon inhibition of TOR. This site is conserved in yeast, and is shown to be oxidized with H_2_O_2_ treatment, suggesting this site is involved in the response to chemical stressors [82].

### 4.3. Carbohydrate Metabolism

The assessment of cell composition after treatment with AZD8055 showed a significant increase in the percentage of carbohydrates as a function of dry mass in the cells beginning 36 h after treatment (Figure 3c). Previous work in plants and a red alga supports this, also showing an increase in starch accumulation with inactivity of TOR [83,84]. This result is further reflected in the changes in thiol oxidation found in this study, with 214 identifiers on 74 proteins showing reversible oxidation, 15 of these identifiers increasing in oxidation and one decreasing (Appendix A). The increase in carbohydrate content is supported more specifically by the oxidative regulation of several important enzymes involved in carbohydrate metabolism. NADP-malate dehydrogenase (Q9FNS5), has one site, Cys389, increasing in oxidation upon treatment (FC: 3.20). This is the chloroplastic isoform of the dehydrogenase, and is well-known to be redox-regulated in higher-order plants [85] and *C. reinhardtii* [86], but this is the first time it has been linked to TOR regulation. This Cys residue is conserved in sorghum and has been shown to be part of a regulatory disulfide bond, with the enzyme being fully active when it is completely reduced [87], suggesting that the increase in oxidation observed is decreasing the overall activity of the dehydrogenase.

The large subunit of isopropylmalate dehydratase (A8JG03) also has a Cys site significantly increasing in oxidation, Cys444 (FC: 2.47). This is a highly modified site, with previous work identifying S-glutathionylation [32], S-nitrosylation [73], and regulation via thioredoxin [74] on this Cys. Although the exact function of this Cys is unknown, it appears to be an important site for oxidative signaling in the cell. 

The identifier with the largest fold change on a carbohydrate-related enzyme is a hypothetical protein (A0A2K3DY10, FC: 20.92) with a sequence almost identical to that of chloroplastic sedoheptulose-1,7-bisphosphatase (P46284), which shares this identifier. This Cys116 site is known to be part of a disulfide bond controlled by thioredoxin, which activates the enzyme upon reduction of the disulfide [88]. With its important role in carbon fixation, the regulation of this enzyme could be contributing significantly to the increase in carbohydrate content that is seen in the cell. 

### 4.4. TOR Pathway-Related Proteins

A total of 28 identifiers on 10 proteins of the known TOR signaling pathway had reversible oxidation identified in this study (Appendix A). Of those identifiers, only one, Cys26 on the TOR complex subunit Lethal with SEC-13 (LST8, A8JDD2), was shown to be significantly changing, with a four-fold increase in oxidation. Although a previous study on mTORC1 has shown that TOR activity is redox regulated [89], this is the first time that an oxidation site has been identified on LST8, which could impact the formation and activity of the TOR complex. While not shown in this dataset, the presence of oxidative sites on the other components of the TORC complex, TOR and Regulatory-associated protein of TOR (RAPTOR), each of which contain several cysteine residues, cannot be excluded; however, more specific enrichment/fractionation methods would need to be used to get the coverage needed for this determination.

A majority of the other TOR-pathway-realted identifiers were from vacuolar ATPases, many of which have been previously shown to be associated with the TOR pathway in mammalian systems [90]. This class of enzymes has high sequence homology, with conserved domains and subunits, and is known to be regulated by Cys oxidation. Although none of these identifiers in this study are changing with TOR inhibition, Cys oxidation was identified on V-type proton ATPase subunits C (A8HYU2), F (A8HZ87), and H (A8HQ97), as well as vacuolar ATP (V-ATP) synthase subunits A (A8I164), B (A8IA45), and E (A8IW47). Interestingly, Cys247 on V-ATP synthase subunit A is a conserved Cys shown to modulate its activity in Arabidopsis [91]. Although not changing, oxidation on this subunit suggests that this site may have a similar mechanism in *C. reinhardtii*, but is not regulated by TOR under the presented conditions. However, Subunit B of V-ATP synthase has been previously shown to be regulated by TOR via phosphorylation on Ser8 [16]. With ATPases’ important role of maintaining cell homeostasis under stress, this subunit could be an important component of TOR’s regulatory pathway, but further studies would be required to assess its exact role.

### 4.5. Photosynthesis

There were 20 photosynthesis-related proteins with 21 significantly changing reversible oxidation identifiers (Appendix A), suggesting that inhibition TOR plays a role in regulating the light reactions of photosynthesis, a novel finding. Looking specifically at the protein components of photosynthetic machinery, 13 proteins were found to have oxidation on Cys with a total of 60 identifiers (Figure 6). Two of these proteins, photosystem I iron-sulfur center (PsaC, Q00914) and Ferredoxin—NADP reductase (FNR, A8J6Y8) have an identifier significantly increasing in oxidation (FC: 2.46 and 3.80, respectively) and two proteins, Cytochrome f (PetA, P23577) and Ferredoxin (PetF, A8IV40), have identifiers significantly decreasing in oxidation (FC: 0.39 and 0.36, respectively). OJIP analysis suggests that TOR inhibition results in decreased electron flow through PSII, likely stemming from an overly reduced plastoquinone pool due to downstream effects, resulting in decreased overall turnover of PSII and linear (oxygen-producing) photosynthetic activity. This diminished activity is irreversible and begins within 15 min of inhibition and continues through the full 48 h of treatment. While the lack of chlorosis (degradation of chlorophyll) suggests the photosynthetic apparatus is still intact, these data suggests that TOR inhibition results in a marked inhibition of electron flux through PSII. Furthermore, the increased reduction of the plastoquinone pool downstream of PSII suggests that either downstream proteins of PSII have been damaged or that PSI is participating in cyclic, rather than linear, electron transport. Cyclic electron transport results in greater production of ATP as FNR is bypassed, resulting in electron transfer to cytochrome *b*_6_*f* via the stromal side, increasing the proton gradient needed for ATP synthesis while decreasing the production of NADPH. This leads to a more reduced quinone pool, as fewer electrons are transferred to CO_2_ via NADPH.

A shift to cyclic electron transport is supported by proteomic analysis, as there is a 3.8x FC in oxidation of FNR (A8J6Y8), showing a decrease in enzyme activity (FNR is reduced by ferredoxin following reduction by PSI) [92]. If FNR was generating NADPH, an increase in reduction, not oxidation, would be expected, as it would be reduced in the process of shuttling electrons. A shift toward cyclic electron transport is further indicated by the pairing of decreased PSII activity with the decrease in oxidation of Cys52-Cys55 of cytochrome *f* (P23577, FC: 0.39), the subunit of cytochrome *b*_6_*f* responsible for electron transfer to plastocyanin. An increase in the reduction of cytochrome *f* suggests an increased electron load; while the *b*_6_*f* complex is a rate-limiting step in both linear and cyclic electron transport [93], it would likely be more substantial under cyclic electron transport as electrons are supplied from both plastoquinol and ferredoxin. However, when paired with the increased oxidation of other regulatory Cys, such as PsaC (Q00914, FC: 3.36), a PSI subunit containing one of the iron-sulfur centers, it is also possible that damages downstream of PSII are causing a “traffic jam” for electrons, resulting in a more reduced plastoquinone pool that decreases the available plastoquinone for PSII. It could possibly be a combination of these two phenomena, wherein the ETC converts to cyclic electron transfer due to stress, while simultaneously decreasing capacity due to oxidative changes downstream of PSII. 

## 5. Conclusions

While TOR is a known master regulator with significant control over nutrient-responsive pathways, its role in the metabolic regulation of photosynthetic eukaryotes is still not completely understood. By characterizing the physiological effects of AZD8055-mediated TOR inhibition on *C. reinhardtii* and pairing it with label-free quantitative proteomics following OxRAC, a network of reversible thiol oxidation was unveiled. This complex oxidation network was cell-wide, overlapping all major metabolic processes and indicating an essential role for thiol oxidative signaling in TOR regulation. TOR targets for thiol oxidation included important lipases involved in lipid cycling and TAG biosynthesis, directly linking oxidative signaling to upregulation of TAGs following TOR inhibition. Additionally, for the first time, photosynthesis was shown to be regulated by the TOR pathway with inhibition of TOR causing a decrease in the photosynthetic efficiency of PSII, and a shift toward cyclic electron transport. Future studies will benefit from modification-specific redox analysis, through which the individual regulatory mechanisms could be determined. 

## Figures and Tables

**Figure 1 cells-08-01171-f001:**
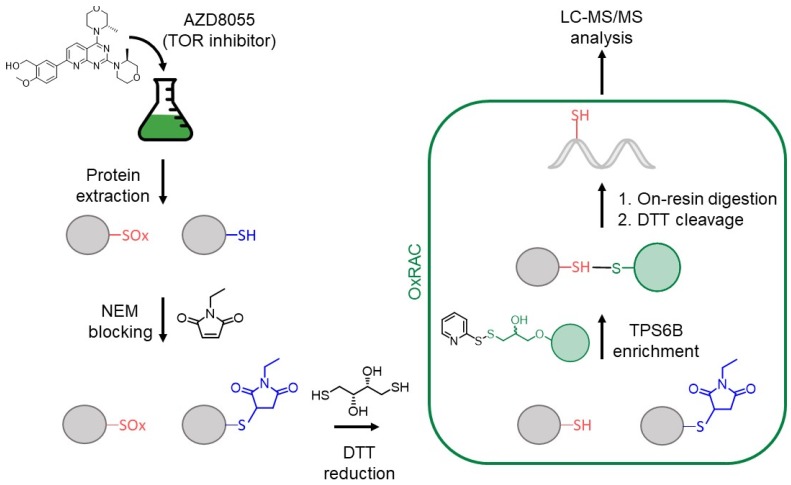
Workflow for proteomic oxidative cysteine analysis of *C. reinhardtii* with AZD8055 treatment. After protein extraction, reduced cysteine thiols are blocked with *N*-ethylmalemide (NEM), before reversibly oxidized cysteines are reduced using dithiothreitol (DTT). An oxidized cysteine resin-assisted capture method (OxRAC) is used to enrich proteins containing oxidized cysteines and samples are processed for bottom-up liquid chromatography—tandem mass spectrometry (LC-MS/MS) analysis.

**Figure 2 cells-08-01171-f002:**
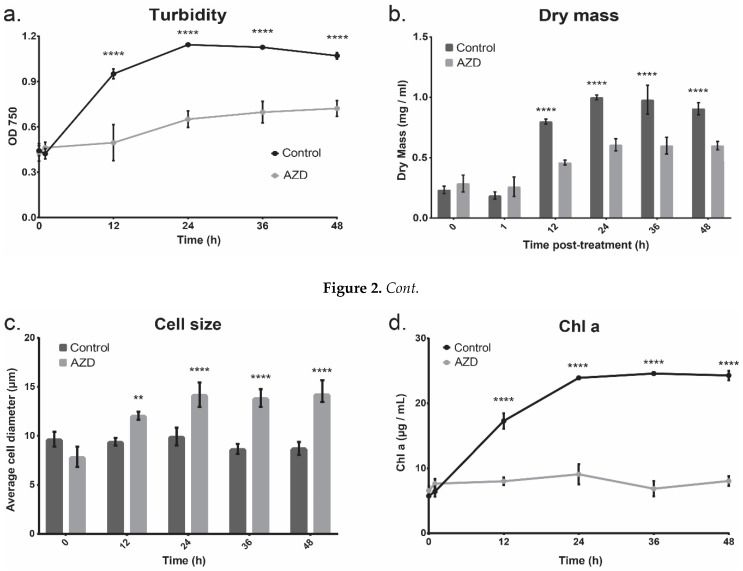
The growth curves of cultures with and without AZD8055 exposure. The error bars represent standard deviation and statistical differences indicate a difference between the inhibited and non-inhibited cultures at one time point. Significance is denoted by asterisks, where *** indicates *p* ≤ 0.001, and **** indicates *p* ≤ 0.0001. (**a**) The turbidity (optical density) of *C. reinhardtii* following dosing with AZD8055. Cells were dosed in mid-exponential phase. Control cultures were dosed with DMSO, the solvent used for AZD8055. (**b**) The total dry mass of the cultures with and without AZD8055 treatment. (**c**) The cell diameter of the cultures with and without AZD8055 treatment. (**d**.) The Chl *a* content of cultures following treatment in mid-exponential phase. Chl *a* is an indication of both organismal health as well as photosynthetic productivity.

**Figure 3 cells-08-01171-f003:**
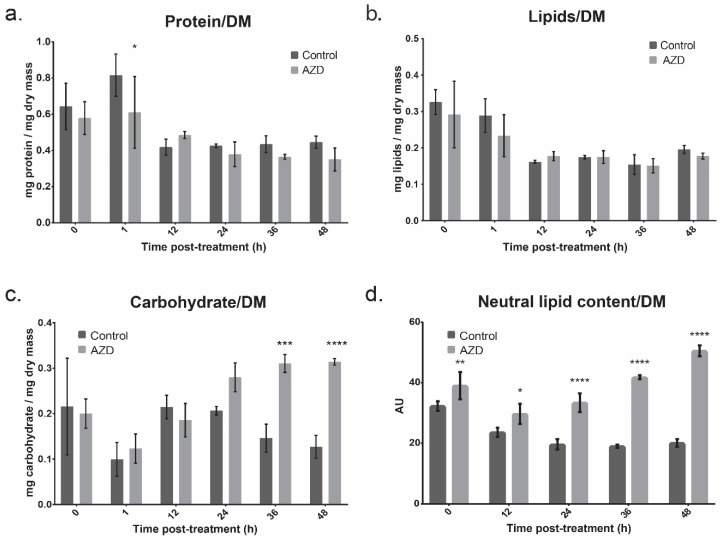
Compositional analysis of *C. reinhardtii* cultures with and without AZD8055 treatment, taken over time. The error bars represent standard deviation and statistical differences indicate a difference between the inhibited and non-inhibited cultures at one time point. Significance is denoted by asterisks, where * indicates *p* ≤ 0.05, ** indicates *p* ≤ 0.01, *** indicates *p* ≤ 0.001, and **** indicates *p* ≤ 0.0001. (**a**) The total protein content of the cultures, measured in mg/mg dry mass, with and without AZD8055 treatment. (**b**) The total lipid content of cultures, measured in mg/mg dry mass. (**c**) The total carbohydrate content of cultures, measured in mg/mg dry mass. (**d**.) Neutral lipid content of the cultures with and without AZD8055 treatment measured using Nile Red staining.

**Figure 4 cells-08-01171-f004:**
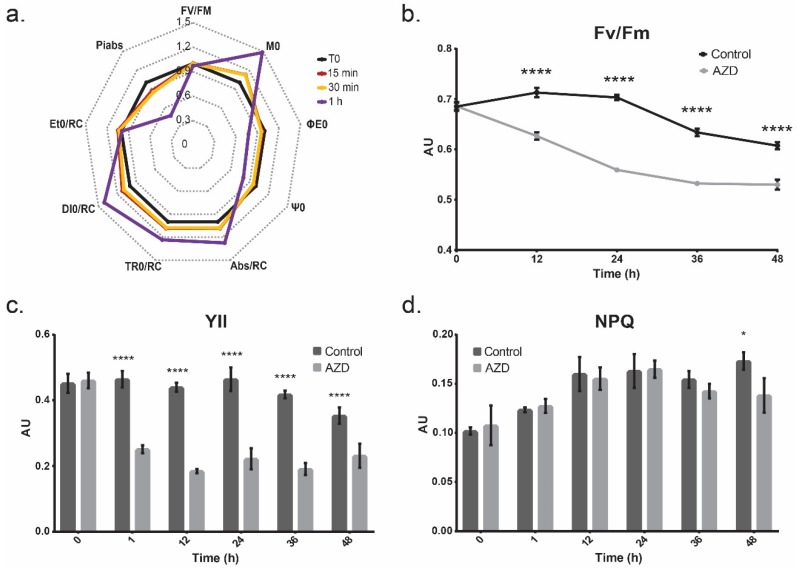
Photosynthesis measurements taken after treatment. The error bars represent standard deviation and statistical differences indicate a difference between the inhibited and non-inhibited cultures at one time point. Significance is denoted by asterisks, where * indicates *p* ≤ 0.05, and **** indicates *p* ≤ 0.0001. (**a**.) Chl *a* fluorescence OJIP parameters of AZD-dosed cultures over the course of 1 h, where t0 is normalized to 1, enabling visualization of rapid changes in the photosynthetic apparatus. Changes in OJIP parameters relative to the control, as well as the derivations and explanations of all parameters, can be found in Appendix A. (**b**.) F_V_/F_M_, the measure of quantum efficiency of PSII following dark adaptation, of the cultures with and without AZD treatment. (**c**.) The photochemical yields of photosystem II with and without treatment with AZD8055 measured using PAM fluorescence. (**d**.) The nonphotochemical quenching of the cultures with and without AZD8055 treatment measured using PAM fluorescence.

**Figure 5 cells-08-01171-f005:**
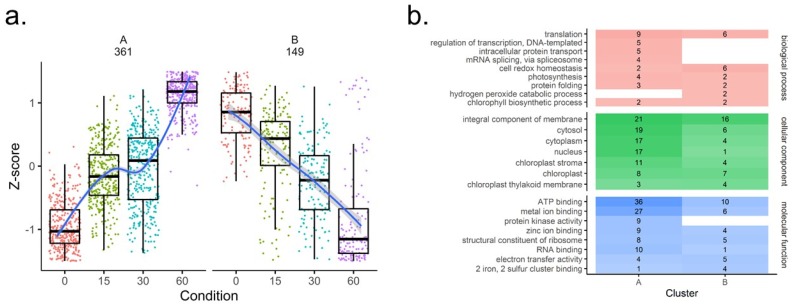
Differential analysis of the reversibly oxidized cysteine thiol proteome. (**a**.) Hierarchical clustering of the 510 identifiers significantly changing (*p* < 0.05, FC > ±2) into two clusters. (**b**.) Gene ontology (GO) summary of significantly changing identifiers in clusters A and B from hierarchical clustering analysis. The number and shading correspond to the number of unique proteins in each category for each cluster.

**Figure 6 cells-08-01171-f006:**
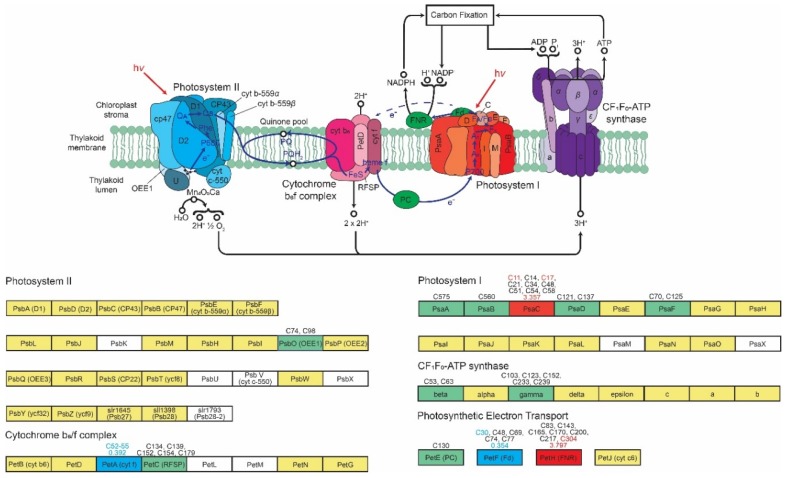
Reversible oxidation on photosynthetic machinery. Adapted from KEGG pathway map for photosynthesis (https://www.genome.jp/dbget-bin/www_bget?pathway:map00195). Protein names are labeled in diagram with gene names listed in boxes below. Components with *C. reinhardtii* homologs are in yellow. Proteins with identified reversible oxidation sites are in green with the Cys sites identified listed above. Proteins with significantly increasing (red) or decreasing (blue) identifiers upon inhibition of TOR also include the maximum fold change listed above.

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
