# Peer review of "Inhibition of TOR in Chlamydomonas reinhardtii Leads to Rapid Cysteine Oxidation Reflecting Sustained Physiological Changes"

_cells, 2019, doi:10.3390/cells8101171_

Round 1

Reviewer 1 Report

This study reports the first quantitative analysis of the redox proteome of a photosynthetic organism in response to TOR inhibition. The authors used the TOR inhibitor AZD8055 to inhibit TORC1 signaling in the unicellular green alga Chlamydomonas reinhardtii. The results indicate that, in addition to inhibiting cell growth, this drug had a negative impact on photosynthesis, which is a novel finding in the field. The authors have performed a detailed quantitative proteomic analysis of reversible thiol oxidation in cells treated with AZD8055 at different times using oxidized cysteine resin-assisted capture enrichment. This analysis will be highly valuable for many groups working in the TOR field as it provides the first study about redox changes that take place in the proteome of TOR-inhibited cells. Notably, some of the proteins undergoing redox modification identified in this study play an important role in photosynthesis, providing a possible mechanism for the negative effect of TOR inhibition on the photosynthetic electron transport chain reported in this study.

Overall, the manuscript is very well written, experiments are well performed and results are consistent. I have only a few suggestions and comments:

In my opinion, the link that authors established between TAG accumulation, oxidative stress and TOR signaling is somehow forced. It has been shown that TOR inhibition raises the TAG content in Chlamydomonas and also that TAGs accumulate in response to oxidative stress, but this does not necessarily imply TOR is regulated by oxidative signals (as stated in the Abstract, lines 17-19). The quantitative redox proteomic analysis performed in this study is important enough and does not need to be “justified” by the common effect of TOR inhibition and oxidative stress on TAG accumulation. I would change this sentence from the abstract.

Lines 430-431: references to the studies showing TOR inhibition within minutes should be indicated.

Discussion: the identification of vacuolar ATPases as redox targets in cells treated with AZD8055 is interesting as TOR inhibition in Chlamydomonas leads to increased vacuolization and autophagy activation. Perhaps authors could discuss about the important role that vacuolar ATPases may play downstream of TOR.

LST8 has been identified in this study as one of the proteins that significantly changes its oxidation state. This interesting finding suggests that this modification might be relevant for TOR function. Besides LST8, TORC1 is composed by the TOR kinase and raptor proteins and both of them are large proteins that contain several Cys residues. Can the authors exclude that TOR and raptor do not undergo reversible oxidation?

Line 449: typo (“which we has been shown”).

Reviewer 2 Report

The authors compared AZD8055-treated Chlamydomonas cells and th e control cells in cell growth, syntheses of macromolecules (protein, lipid, carbohyfrate etc.), photosynthetic activity, and Cys-oxdized proteins. They congregate these results together to conclude that TOR inhibition induces cellular oxidative conditions which changes metabolic pathway such as accumulation of triacylglyceol.

However, this reviewer thinks that these data are too preliminary for the authors' conclusion. For example, TOR (Tor complex1) has a multiple role in protein synthesis, and TOR inhibition affects biogenesis of many species of macromolecules shown in Fig.2b. But the authors only focused on an oxidative effect of TOR inhibition and ignored another effects. They should test various conditions (e.g. another protein synthesis inhibitors, oxidative stress, dark and starvation conditions). Therefore, this reviewer does not recommend this manuscript to be published in Cells Journal.     

Reviewer 3 Report

This is a report on the importance of TOR signaling pathway based on growth pattern of Chlamydomonas reinhardtii, changes in macromolecule composition, photosynthetic measurements, and changes in the reversibly oxidized cysteine thiol proteome after treatment with AZD8055. Authors interpreted that the global oxidative changes caused by TOR inhibition mirrored physiological modifications examined in this manuscript. I think that this manuscript demonstrates well that TOR plays a central role in regulating many aspects of cellular physiological states including photosynthesis through reversible thiol oxidation. Especially, authors showed the extensive analyses in processes of photosynthesis, some of which may be targets of TOR signaling. They provided the proteomic changes of redox-controlled proteins, which seemed to be further developed from their previous study and now were applied to TOR signaling pathway. Nevertheless, I feel that this study is preliminary to be published in the following senses and I hope authors to consider them seriously for contributing to communities studying the TOR signaling.

Major comments

Although this manuscript delivers many valuable analyses to readers, in my opinion, data do not clearly show how TOR signaling affects the many aspects of cellular physiology including photosynthesis. When TOR is inhibited by any means, it is quite expected that there will be serious changes or damages many of which may eventually lead to autophagy. In this sense, drastic changes in such and such physiology in Figures are expected although diverse techniques were used and extensive hypothetical description on how was made in Discussion. I think authors should have made efforts in answering the following questions and providing substantial data in, at least, a few cases that authors mentioned in Discussion.

-Which components of TOR signaling actually affect the physiological changes?

-Which specific step(s) of each metabolism or photosynthesis was(were) the target(s) of TOR signaling?

Author mentioned the “direct” inhibition of photosynthesis in response to TOR inhibition (line 76). Does TOR kinase (or its downstream, e.g., S6K1) directly act on any step(s) of photosynthesis? Or did TOR inhibition result in initiating autophagy, minimizing the energy-consuming processes such as photosynthesis? I think authors need to provide more solid evidence on the “direct” connection between TOR signaling and photosynthesis to keep such comments.

Many other studies of TOR inhibition by diverse means other than AZD8055 have been published. There needs to be comparison between results in this manuscript and other TOR inhibition studies, and the some critical points of the comparsion needs to be mentioned. I hope there will be a few make-up experiments after authors went through the comparison, such as examining the level of gene expression or the protein for the deficient steps in AZD-treated algae authors mentioned.

Minor comments

Abbreviations in Figure 4 need to be properly defined. Figure legends to Suppl Figures are not found.

Round 2

Reviewer 2 Report

The authors did not perform any experiment in response to this reviewer. If they want to accept their manuscript in the present form, please ask the editor to change reviewer. 

Reviewer 3 Report

Authors have modified the manuscript in some part in response to my comments, while they expressed their different opinion from one of major comments. Considering the reputation of this journal in this field, I think this manuscript is not ready to be published although many valuable analyses and interesting perspectives are presented.

     Presenting the direct targets of TOR is not an absolute requisite for this manuscript. Although authors found out global changes in redox-controlled proteins by TOR inhibition, they do not have biological meaning yet if the data really do not show close connectivity to TOR signaling components. Authors need to show substantial data on which components of TOR signaling may affect the changes they found, or which step(s) of each metabolism or photosynthesis was (may be) targeted by TOR signaling. I found no data to show such connectivity. In my opinion, this manuscript shows inhibition-and-effects, but not "HOW" in between. Authors, at least, should present data to show some possible connectivity. As authors found possible targets of TOR in their previous study, they could pick up some of those (or some from previously known TOR targets) and try to examine possible connection of them to TOR signaling. If there is no such information presented, I think this manuscript is not ready yet to be published although data in this manuscript are valuable and interesting.

Author’s responses to other comments are understandable.